

**Freeze-thaw processes correspond to the protection-loss of soil**
**organic carbon through regulating pore structure of aggregates**
**in alpine ecosystems**
Ruizhe Wang[1,2], Xia Hu[1,2*]
1 State Key Laboratory of Earth Surface Processes and Resource Ecology, Faculty of Geographical Science, Beijing
Normal University, Beijing 100875, China.
2 School of Natural Resources, Faculty of Geographical Science, Beijing Normal University, Beijing 100875, China.
*Correspondence* to: Xia Hu (huxia@bnu.edu.cn, +86-010-58800238)





**Abstract.** Seasonal freeze–thaw (FT) processes alter soil formation and causes changes in soil structure in alpine
ecosystems. Soil aggregates are basic soil structural units and play a crucial role in soil organic carbon (SOC)
protection and microbial habitation. However, the impact of seasonal FT processes on pore structure and its impact
on SOC fractions have been overlooked. This study characterized the pore structure and SOC fractions of aggregates
during the unstable freezing period (UFP), stable frozen period (SFP), unstable thawing period (UTP) and stable
thawed period (STP) in typical alpine ecosystems via the dry sieving procedure, X-ray computed tomography (CT)
scanning and elemental analysis. The results showed that pore characteristics of 0.25-2 mm aggregates were more
vulnerable to seasonal FT processes than that of > 2 mm aggregates. The freezing process promoted the formation of >
80 µm pores of aggregates. The total organic carbon (TOC), particulate organic carbon (POC) and mineral-associated
organic carbon (MAOC) contents of macroaggregates were high in the stable frozen period and low in unstable
thawing period, demonstrating that freezing process enhanced SOC accumulation while early stage of thawing led to
SOC loss. The vertical distribution of SOC of aggregates was more uniform in stable frozen period than in other
periods. Pore equivalent diameter was the most important structural characteristic influencing SOC contents of
aggregates. In the freezing period, the importance of pore structure in regulating SOC protection was more obvious
and pore structure inhibited SOC loss by promoted the formation of >80 µm pores. In the thawing period, pores of 15-
30 µm inhibited SOC protection. Our results are valuable for evaluating potential changes in alpine soil carbon sinks
under global warming.
**Key words: Seasonal freeze–thaw process, soil aggregate, soil organic carbon, soil pore**





## 1. Introduction

Freeze–thaw (FT) cycles are main process of soil formation in alpine regions and more than half of the Northern Hemisphere's soil is affected by FT cycles (Wang et al., 2007). Ongoing global warming has reduced snow cover in winter and decreased the insulations of soils against freezing, which has increased the frequency of FT cycles (Kreyling et al., 2008). The expansion of water volume with freezing and the shrinkage after thawing destroys soil aggregates and soil organic carbon (SOC) protection (Oztas and Fayetorbay, 2003; Tan et al., 2014). FT processes not only affect the stability of soil aggregates but also change their inner pore characteristics, especially those of the water-filled pores (Wang et al., 2012; Li and Fan, 2014; Starkloff et al., 2017). A decrease in pore connectivity and an increase in elongated porosity were observed after continuous FT events (Ma et al., 2020; Rooney et al., 2022). FT cycles could lead to opposite changes in soil porosity of exterior aggregates and interior aggregates (Zhao and Hu, 2023a). FT processes have profound effects on the carbon cycles, accelerating the loss and decomposition of SOC in the soil carbon pool of terrestrial ecosystems (Yi et al. 2015). FT processes can change SOC distribution by stimulating substrate release (Song et al., 2017), destroying soil aggregates and affecting microbial activities (Campbell et al., 2014; Xiao et al., 2019). The impact of FT processes on SOC fraction contents varies due to the differences in their formation pathways and physio-chemical stabilities. FT processes could significantly increase soil soluble carbon content and extractable SOC content but decrease microbial biomass carbon (MBC) (Patel et al., 2021). However, these related studies were mostly based on simulated indoor freeze–thaw experiments. The differences between simulated FTC conditions and field FT conditions are in that (1) the indoor FTC cannot simulate the dynamics of vegetation growth, which had significant impacts on SOC origin and soil structure and (2) the indoor FTC cannot reflect the changes in environmental factors which impacts the FT conditions. Quantifying pore structure and SOC fractions of soil aggregates during the seasonal FT process is valuable for understanding how freezing and thawing affect soil structure and functions.

Soil structure refers to the spatial arrangement of solids and voids and controls many important biophysical processes in soils (Rabot et al., 2018). Soil aggregates are basic units of soil and are important indicators of soil structure and soil quality (Six et al., 2000; 2004). According to their diameters, soil aggregates can be divided into microaggregates (< 0.25 mm) and macroaggregates (> 0.25 mm). Macroaggregates play a key role in stabilizing SOC, maintaining levels of water and nutrients and microbial habitation (Ananyeva et al., 2013; Chen et al., 2019; Wang et al., 2022). SOC is preserved by physical protection in the forms of light organic carbon (fLOC), particulate organic carbon (POC) and mineral-associated organic carbon (MAOC). POC is a crucial contributor to soil aggregation and parallels plant-derived carbon into aggregates, and MAOC plays a crucial role in long-term SOC storage (Wang et al.,





2020; Witzgall et al., 2021). This classification benefits a more extensive understanding of soil carbon maintenance
and response to climate change and land use (Kallenbach et al., 2016; Lavallee et al., 2020; Liao et al., 2023). The
pore networks of soil aggregates are heterogeneous. They can directly influence the accessibility of organic matter to
microbes and indirectly influence microbial activities, thus determining the magnitude to which the SOC is protected
(Ruamps et al., 2013; Kravchenko and Guber, 2018). Interactions between pore structure and SOC soil aggregates
have gained much attention. Zhang et al. (2023) proposed that pore structure alone explained 6.41% and 12.64% of
the variation in the SOC of aggregates in the topsoil and subsoil, respectively. Pores of 30-75 μm   and > 13 μm in
size were found to enhance the mineralization of carbon (Lugato et al., 2009; Kravchenko et al., 2015). Pores of > 90
μm (Quigley and Kravchenko., 2022) and < 15 μm in size (Ananyeva et al., 2013) were found to support SOC
protection. 30–150 μm pores are also the preferential places for new carbon inputs and greater abundance of such
pores translates into a higher spatial footprint that microbes make on SOC storage capacity (Kravchenko et al., 2019).
Therefore, the relationships between pore structure and SOC varied significantly under different soil conditions. Soils
in alpine ecosystems store large amounts of SOC and microorganisms habiting in alpine ecosystems were found to be
more tolerant to cold environments than those in other ecosystems (Zhao and Hu, 2023b). These results indicated that
relationships between pore structure and SOC fractions of alpine soils can be complex and are critical for
understanding the microscale mechanisms of the alpine carbon sinks. The dynamics of SOC during the FT process
could be significantly correlated with the transformation and destruction of aggregates (Dagesse, 2013). However, the
relationships between pore structure and SOC fractions are still not well understood for alpine soils under FT processes.

The Qinghai-Tibet Plateau (QTP) has the largest permafrost coverage in the middle to low latitudes of the world

(Wu et al., 2010). Soils of the QTP are fragile and vulnerable to environmental changes. Dramatic changes have
occurred in FT occurrence in recent years in the QTP as the depth and duration of FT processes have decreased while
the frequency of FT cycles has increased (Peng et al., 2017). Previous studies have demonstrated that FT cycles are
critical in influencing soil structure of the QTP (Gao et al., 2020; Yang et al., 2021). Alpine meadow soil
macroaggregates of the QTP had dense pore networks with many elongated pores in them due to frequent FT cycles
(Wang and Hu, 2023). FT processes led to opposite changes in the intra-aggregate porosity and interaggregate porosity
of alpine meadow soil (Zhao and Hu, 2022). Soils of the QTP serve as important SOC pools, and 36%-50% of the
SOC is concentrated in the 0-50 cm soil layer (Ding et al., 2016; Mu et al., 2020). Intensified FT cycles, especially
thawing processes, have led to large amounts of SOC loss, while the changes in SOC fractions remain unknown (Todd-
Brown et al., 2014; Liu et al., 2018). Liu et al. (2022) reported that topsoil POC content decreased upon permafrost
thawing, while MAOC content remained stable on the QTP. Deeper soil layers are more active in SOC changes as



these layers spend more time thawing (Chen et al., 2023). However, the responses of pore structure and SOC fractions
of aggregates to seasonal FT processes have been overlooked due to difficulties associated with monitoring on the
QTP, which has important implications for predicting carbon turnover projections under global warming (He et al.,

2021).

To fill these research gaps, the objectives of the study were: (1) to quantify changes in pore structure and the

SOC fraction content of aggregates in typical alpine ecosystems during the seasonal FT process; (2) to investigate the
relationships between them and (3) to clarify the role of pore structure on aggregate functions related to SOC
protection during seasonal freeze–thaw processes.
**2.   Materials and methods**
*2.1   study sites and sampling*

The study was carried out in the Qinghai Lake Watershed (36°15′N-38°20′N, 97°50′-101°20′E), northeastern QTP.

The area lies in the cold and high-altitude climate zone, with a mean annual temperature and precipitation of 0.1 °C
and 400 mm, respectively (Li et al., 2018). Two ecosystems were selected in the study: *Kobresia pygmaea* meadow
(KPM) and *Potentilla fruticosa* shrubland (PFS). They are representative terrestrial ecosystems of the Qinghai Lake
watershed and account for over 60% of the watershed land area (Hu et al., 2016). One of the main features of these
two ecosystems is the mattic epipedon present on the soil surface. Mattic epipedon is the surface layer consisting of a
grass felt-like complex formed by the interweaving of live and dead roots of different ages. The layer is soft and
significantly affects SOC storage and soil structure (Hu et al., 2023). The soil type was classified as Gelic Cambisols
according to the FAO UNESCO system (IUSS Working Group WRB, 2022). We tried to avoid the simple pseudo
replication so that each sampling sites have a certain distance with others (> 1 km). Three sites within each ecosystem
have similar vegetation conditions. In every FT period, three sampling plots (1 m × 1 m) were set up at each site.



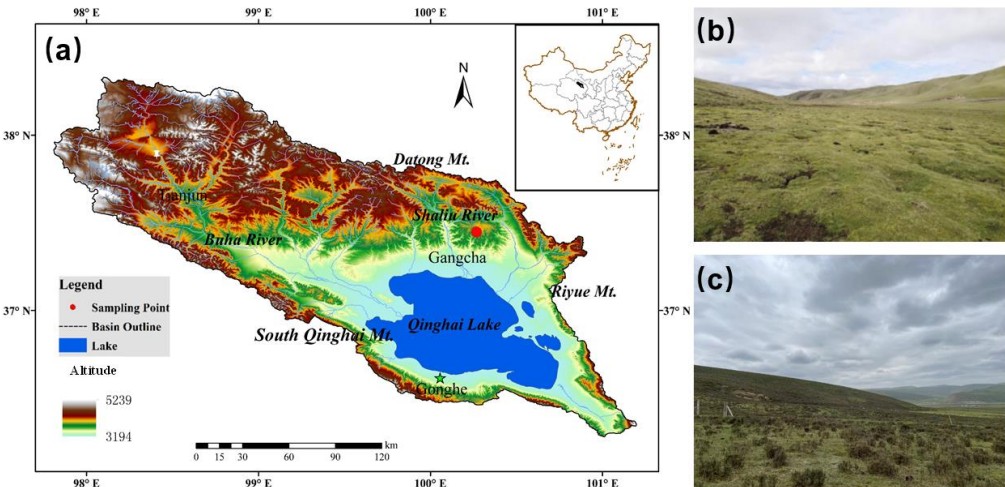


Fig. 1. Location of the sampling site (a) and landscapes of the (b) *Kobresia pygmaea* meadow    ecosystem and (c)
*Potentilla fruticosa* shrub ecosystem.

The division of seasonal FT periods is based on changes in daily soil temperature (Chen et al., 2021; Wu et al.,
2023). The EM-50 soil temperature data for 2019, 2020, and 2021 were obtained at 0.5 Hz with 30 min averages at
all three study sites using the ECH2O 5TE sensor (Decagon Devices, USA) (Li et al., 2018). The seasonal freeze–
thaw process was divided into four periods in this study: the unstable freezing period (UFP, as soil temperature starts
to drop to 0℃), the stable frozen period (SFP, with soil temperature completely blow 0 ℃), the unstable thawing
period (UTP, as soil temperature starts to rise above 0 ℃), and the stable thawed period (STP, with soil temperature
completely above 0 ℃). The freezing process included the SFP and UFP, while the thawing process included the STP
and UTP. Soil samples were taken in October 2021 (representing UFP), January 2022 (representing SFP), May 2022
(representing UFP) and July 2022 (representing SFP).



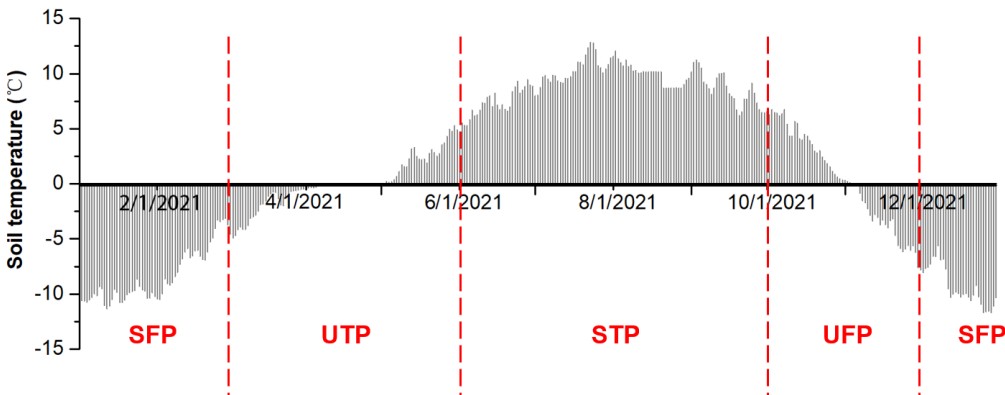


Fig. 2. Daily average soil temperature in 2021 and the classification of freeze–thaw stages (SFP-stable frozen period,
UTP-unstable thawing period, STP-stable thawing period and UFP-unstable freezing period).

In every site, soils samples from three soil profiles were dug for replicates. A total of 18 soil profiles were obtained
in every FT period. We classified the soil layers as 0-10 cm, 10-30 cm and 30-50 cm soil layers. Soil cores and bulk
soil were collected at each soil layer for aggregate sieving and physiochemical characteristic measurements,
respectively. Soil cores were obtained using an 80 mm diameter soil auger and then preserved in an icebox before
being sieved in the laboratory. A total of 54 soil cores were collected in every FT period. The basic soil properties of
each soil layer at the study site are listed in Table S1. Particle size distribution was determined using the sieve-pipette
method (Mako et al., 2019; Zhao et al., 2021). The soil water content as weight was determined using an oven-dried
method (Klute, 1986). Soil pH measurements were conducted by an FE20 pH meter (Mettler Toledo, Columbus, USA)
from slurries of samples at a soil:water ratio of 1:2.5 (w:w) (Zhao et al., 2020). SOC and TN were determined using
a CN 802 elemental analyzer (VELP, Italy). Inorganic carbon was removed from the soil samples using 1 mol/L HCl
prior to elemental analysis (Zhang et al., 2017).
*2.2  Aggregate sieving*
Separation of the soil aggregates was performed using the dry sieving method with 0.053, 0.25- and 2-mm sieves
from top to bottom. Soil cores were gently broken by hand into 1-cm clods, and then soils were laid out between sheets
of brown paper (Schutter and Dick, 2002). Debris such as gravel and roots were removed from the samples. Two
hundred grams of soil was placed on the top sieve and was shaken for five minutes by the sieve shaker. Therefore, the
aggregates were divided into four categories: large macroaggregates (LMAs, with diameters >2 mm), small
macroaggregates (SMAs, with diameters of 0.25~2 mm), microaggregates (mAs, with diameters of 0.053~0.25 mm),
and fractions with diameters <0.053 mm. aggregate fractions of LMAs and SMAs were weighed and preserved for



further analysis.
*2.3  CT scanning and image processing*

A nanoVoxel-4000 X-ray three-dimensional microscopic CT (Sanying Precision Instruments Co., Ltd., China)

was used to scan the soil aggregates with X-ray source parameters of voltage 80 kV and current 50 µA, with which
2800 detailed and low-noise images could be obtained during a 360° rotation. The reconstructed images featured a 3.6
µm spatial resolution and 2800 × 2800 × 1500 voxels. Aggregate fractions of > 2 mm and 0.25-2 mm from all soil
layers of the UFP, SFP, UTP and STP periods were scanned (other fractions were too small to separate into a single
sample).

Reconstruction of the pore network of aggregates was completed using Avizo 9.0 (Visualization Sciences Group,

Burlington, MA). The procedure for image analysis was similar to that described by Wang and Hu (2023) (Fig. 3).
Briefly, the clutters around the aggregates were eliminated using a volume-editing module. Mask extraction was
carried out in the segmentation module (Zhao et al. 2020). The soil matrix was selected with the "Magic Wand" tool,
and then the "Fill" tool was used to fill the pores for obtaining the aggregate boundary and the mask of the whole
aggregate (Zhao and Hu, 2023a). All images were binarily segmented using the histogram thresholding method based
on the global thresholding algorithm (Jaques et al., 2021), and pore thresholds were selected for all images.

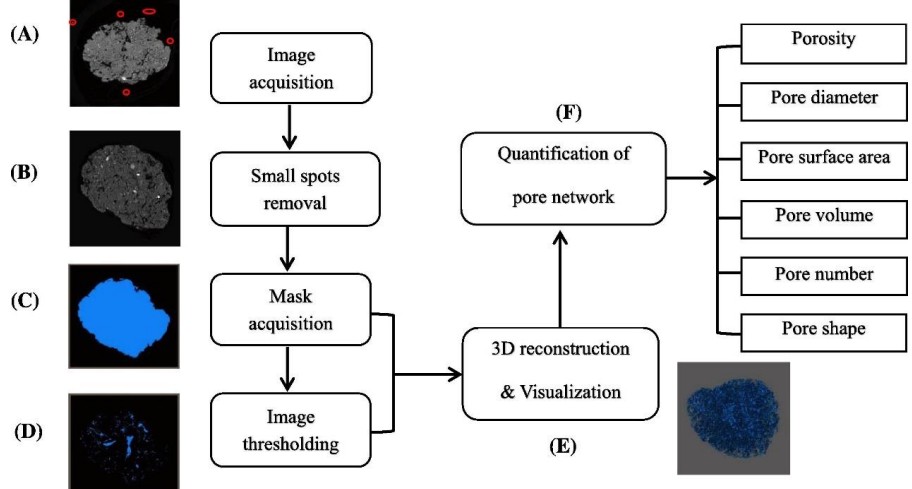


Fig. 3. Procedures used for the visualization and quantification of soil aggregate pore networks. Taken from Zhao et

al. (2020) with permission from Elsevier.


The two-dimensional images were transformed into 3D images by Volume Rendering tool in Avizo 9.0 software.





The intra-aggregate porosity was calculated using the Volume Fraction tool. The Volume Rendering tool transferred
the 2D images into 3D images before the equivalent diameter, volume, number, length, and surface area were
calculated using the Label Analysis tool. The pore number density (ND) is defined as the ratio of the pore number (n)
to the total volume of the aggregate samples (V):

$$ND = \frac{n}{V} \tag{1}$$

One pore network may consist of several branches of connected pores or just one individual pore. The pore length
is the total actual length in all branches. The pore length density (LD) is defined as the ratio of the pore length (L) to
the total volume of pores (V) (Yang et al., 2021):

$$LD = \frac{L}{V} \tag{2}$$

The surface area density (SD) is defined as the ratio of the pore surface area (S) to the total volume of V:

$$SD = \frac{S}{V} \tag{3}$$

To characterize the pore shape, the pore shape factor (SF) was calculated as follows:

$$SF = \frac{A_0}{A} \tag{4}$$

where $A_0$ represents the surface area of the equivalent sphere of the pores and $A$ is the actual surface area of
the pores. SF values closer to 1 indicate a more regular pore shape (i.e., closer to a spherical shape), and smaller values
refer to more irregular or elongated pore shapes (Zhou et al., 2012).
The equivalent diameter (EqD) was defined as the diameter of spherical particle with the same volume and was
calculated by pore volume:

$$EqD = \sqrt[3]{\frac{6 \times V}{\pi}} \tag{5}$$

Where $V$ represents the volume of pores.
The pores were divided into four classes based on their equivalent diameter: <15, 15–30, 30–80, and >80 μm.
According to Lal and Shukla (2004), pores <30, 30–80, and >80 μm are termed micropores, mesopores and
macropores, respectively.
*2.4 SOC fraction separation*
In every FT period, macroaggregate samples were sufficiently ground to pass through a 0.15 mm sieve before
their total organic carbon content (TOC) content was measured using the CN 802 elemental analyzer (VELP, Italy).
The determination of SOC fractions, including POC and MAOC, was performed as described by Cambardella



and Elliott (1992). Approximately 5 g of each dried aggregate of the LMA and SMA fractions was moved to a 50 mL
centrifuge tube and dispersed in 25 mL of a sodium hexametaphosphate (0.5%, w/v) solution by shaking for 18 h in a
reciprocating shaker at 120 RMP to ensure that it was evenly blended. The dispersed samples were rinsed onto a 53
µm sieve to separate MAOC (particle size <53 µm) and POC (particle size >53 µm) using distilled water until the
water stream was clear and free of fine soil particles. After that, samples were transferred to evaporating dishes and
dried at 65 °C for 48 h to isolate soils which contained POC or MAOC fractions solely (Six et al., 1998). After
weighing and sieving, all the fractions' SOC contents were measured using the CN800 elemental analyzer. The POC
and MAOC contents were obtained by multiplying the percentage of each particle size fraction in the soil (Sun et al.,

2023).

*2.5 Statistical analysis*
All statistical analyses except redundancy analysis (RDA) were conducted with IBM's SPSS 20 software (SPSS
Inc., USA). One-way analysis of variance (ANOVA) was conducted to compare differences between the four seasonal
freezing-thawing stages and different aggregate fractions. Pearson's correlations were conducted to evaluate the
linkages between pore characteristics and SOC fractions of macroaggregates. RDA was conducted to determine pore
parameters that had a significant impact on SOC fractions and was carried out in R software (http://www.r-project.org)
using the vegan package.
**3.  Results**
*3.1 Soil pore characteristics of aggregates*
Fig. 4 depicts the pore size distribution of soil aggregates during the seasonal FT process. In the two ecosystems,
pores of > 80 µm dominated the pore space in all periods and accounted for over 65% of the total porosity. The
contribution of pores of < 15 µm was low in the stable frozen period with 4.39 % in the meadow ecosystem and 5.36 %
in the shrubland ecosystem. The volume percentage of pores of > 80 µm was high in the stable frozen period (80.62%
in the meadow ecosystem and 87.65% in the shrubland ecosystem) and was significantly higher than that in the UTP
(74.17% in the meadow ecosystem and 78.53% in the shrubland ecosystem) and the STP (67.18% in the meadow
ecosystem and 80.96% in the shrubland ecosystem) (P<0.05). The results showed that freezing process enhanced the
formation of pores of > 80 µm.





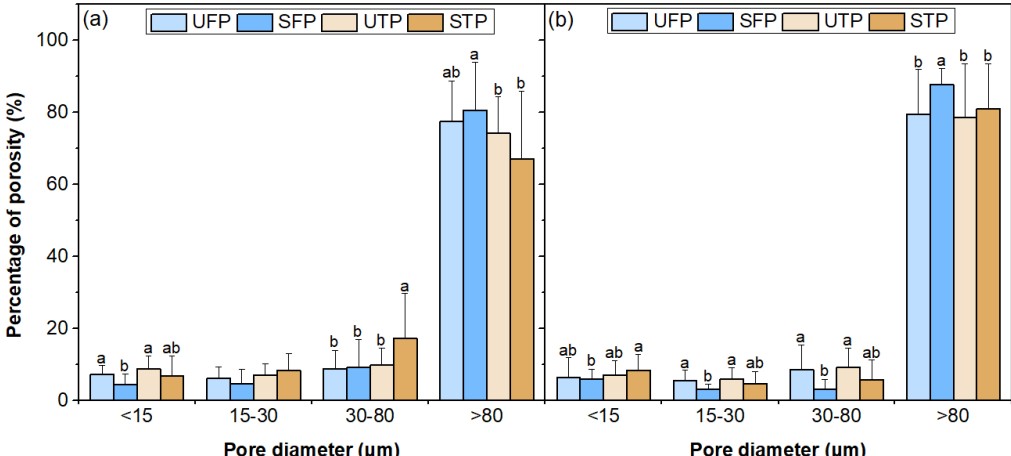

Fig. 4. Pore size distribution (by pore diameter) of soil aggregates in the (a) meadow ecosystem and (b) shrubland ecosystem during the seasonal FT process. Bars represent the mean ± standard error (n=18). Different lowercase letters denote significant differences among pore volume percentages in different FT periods (P<0.05).

Note: UFP-unstable freezing period, SFP-stable frozen period, UTP-unstable thawing period, STP-stable thawed period.

The characteristics of the pores of aggregates during the seasonal FT process are shown in Fig. 5. The seasonal FT process did not significantly affect the EqD (Fig. 5b). The mean pore volumes of 0.25-2 mm aggregates in the freezing period ($3.76 \times 10^3$ μm³ and $3.14 \times 10^3$ μm³ in the meadow and shrubland ecosystems respectively) were significantly higher than those in the thawing period ($2.30 \times 10^3$ μm³ and $2.24 \times 10^3$ μm³ in the meadow and shrubland ecosystems respectively), while no significant difference was observed for > 2 mm aggregates (Fig. 5c). In the meadow ecosystem, the pore length density of the 0.25-2 mm aggregates was $1.68 \times 10^{-4}$ μm μm⁻³ in thawing period, which was 1.71 times higher than that in the freezing period ($0.98 \times 10^{-4}$ μm μm⁻³). In the shrubland ecosystem, pore surface area density and length density of 0.25-2 mm aggregates were $0.0553$ μm² μm⁻³ and $2.37 \times 10^{-4}$ μm μm⁻³, respectively, both significantly higher than those in the freezing period ($0.0404$ μm² μm⁻³ and $1.81 \times 10^{-4}$ μm μm⁻³ for surface area density and length density, respectively). Therefore, seasonal FT processes mainly led to changes in the pore characteristics of 0.25-2 mm aggregates rather than those of > 2 mm aggregates.

In the meadow ecosystem, the SF of pores of the 0.25-2 mm aggregates (0.224 in the freezing period and 0.253 in the thawing period) exceeded those of > 2 mm aggregates (0.164 in the freezing period and 0.184 in the thawing period), while no significant difference in SF was found in the shrubland ecosystem (Fig. 5f).



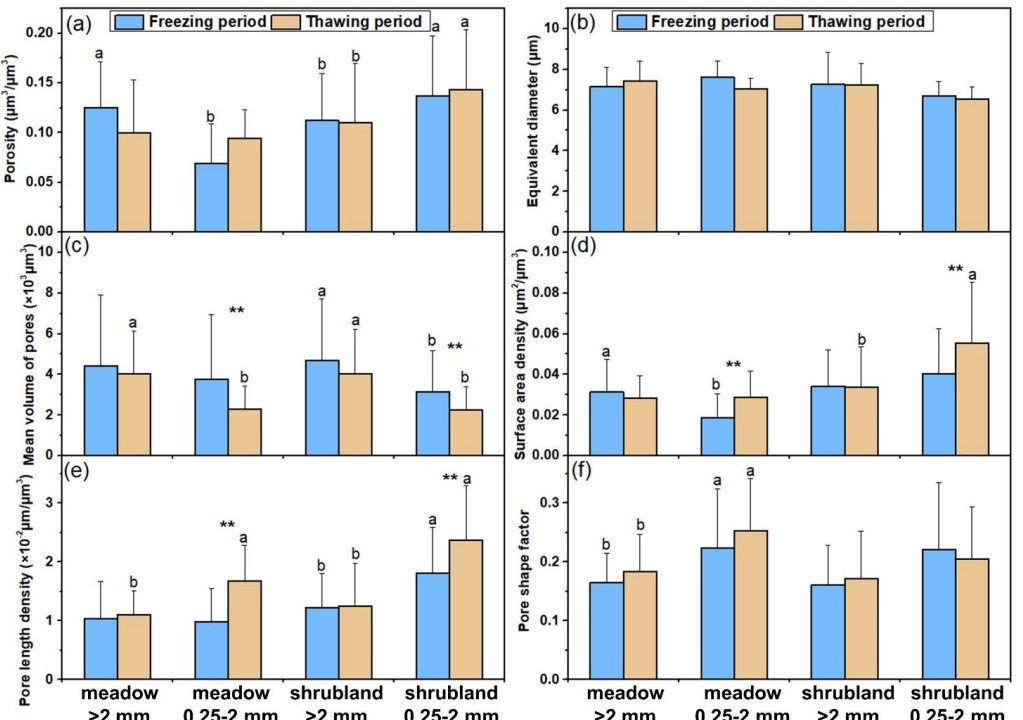

Fig. 5. Pore characteristics of soil aggregates during the seasonal FT process. (a) porosity, (b) pore equivalent diameter, (c) mean volume of pores, (d) pore surface area density, (e) pore length density and (f) pore shape factor. Bars represent the mean ± standard error (n=9). ** represents significant differences between pore characteristics in freezing period and thawing period (P<0.05). Different lowercase letters denote significant differences between pore characteristics of >2 mm aggregates and 0.25-2 mm aggregates (P<0.05).

*3.2 SOC fraction contents of aggregates*

The SOC fraction contents (TOC, POC and MAOC) of aggregates during seasonal freeze–thaw process are shown in Fig. 6. Generally, in the two ecosystems, the TOC contents of aggregates peaked in the stable frozen period, ranging from 57.33 g/kg to 60.28 g/kg (Fig. 6a). The following unstable thawing period demonstrated the significant decline in TOC contents of > 2 mm (dropped by 37.73% and 32.95% in the meadow and shrubland ecosystems, respectively) and 0.25-2 mm aggregates (dropped by 45.57% and 39.43% in the meadow and shrubland ecosystems, respectively) (P<0.05). In the shrubland ecosystem, TOC contents of aggregates in the stable thawed period were also significantly lower than those in the stable frozen period (P<0.05).



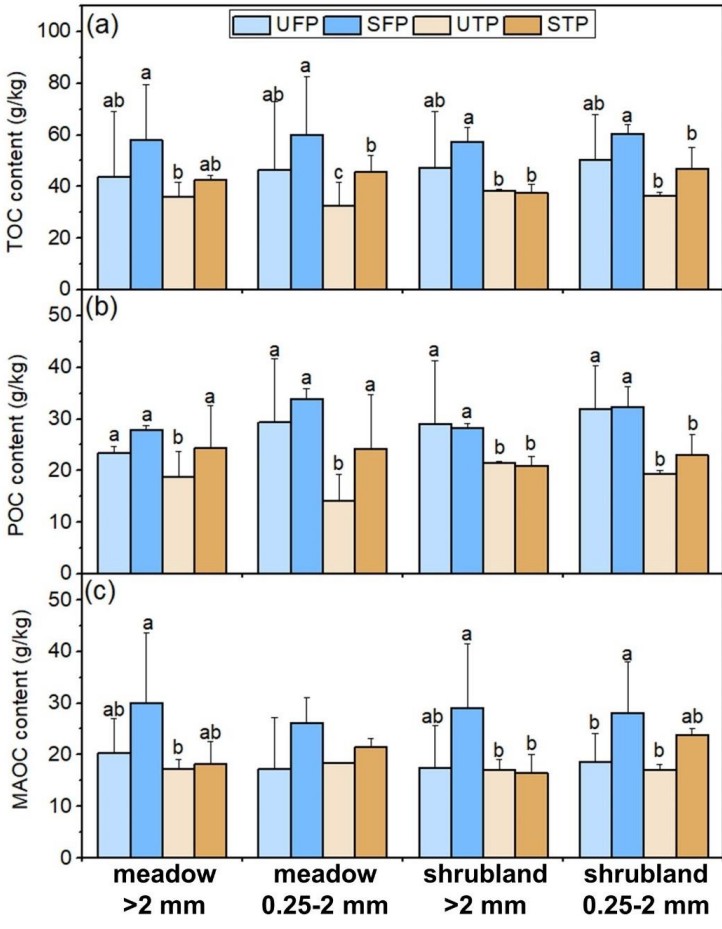

Fig. 6. Changes of SOC content (a-TOC, b-POC and c-MAOC) of soil aggregates during the seasonal freeze–thaw

process. Bars represent the mean ± standard error (n=9). Different lowercase letters denote significant differences

among SOC contents in different FT periods (P<0.05).

Note: UFP-unstable freezing period, SFP-stable frozen period, UTP-unstable thawing period, STP-stable thawed

period.

Changes in contents of POC and MAOC were similar to those of TOC (Fig. 6b and 6c). In the meadow ecosystem,

the POC contents were high in the stable frozen period (27.90 g/kg for > 2 mm aggregates and 33.77 g/kg for 0.25-2

mm aggregates) and the POC contents in the unstable thawing period (18.78 g/kg for > 2 mm aggregates and 14.18

g/kg for 0.25-2 mm aggregates) were significantly lower than those in other periods (P<0.05) (Fig. 6b). The MAOC

content of > 2 mm aggregates was 29.99 g/kg in the stable frozen period, which was 1.74 times higher than that in the





unstable thawing periods (17.28 g/kg) (Fig. 6c). In the shrubland ecosystem, POC contents in freezing periods were
significantly higher than those in thawing periods (P<0.05) (Fig. 6b). The unstable thawing process led to significant
loss in MAOC compared with the stable freezing period (41.54% for POC and 39.14%) (P<0.05) (Fig. 6c).
The changes in the coefficient of variation (CV) during the seasonal FT process, which depicted the variation in
the SOC content of aggregates from different soil depths, were shown in Table 1. In the two ecosystems, the CV values
in the stable frozen period (0.20 for the meadow ecosystem and 0.22 for the shrubland ecosystem) were significantly
lower than those in other periods (P<0.05) . These results revealed that freezing resulted in a more uniform distribution
of SOC across different soil layers.
Table 1 Coefficient of variation (CV) of SOC content of aggregates in all soil layers during the seasonal FT process.

| Ecosystem | Seasonal FT periods | | | |
|---|---|---|---|---|
|  | UFP | SFP | UTP | STP |
| meadow | 0.38a | 0.20b | 0.47a | 0.56a |
| shrubland | 0.46a | 0.22b | 0.34a | 0.34a |

Note: UFP-unstable freezing period, SFP-stable frozen period, UTP-unstable thawing period, STP-stable thawed
period. Different lowercase letters denote significant differences in CV of different FT periods.
*3.3 Relationships between pore structure and SOC fractions of aggregates*
Tables 2 demonstrates the relationships between pore structure and SOC fractions of aggregates in the freezing
process (UFP and SFP) and thawing process (UTP and STP). In the freezing process, the POC content was positively
correlated with pores of <15 μm (P<0.05). The TOC and MAOC contents were both positively correlated with pore
length density (P<0.05). In the thawing process, no correlations were observed between SOC fractions and pore
parameters while pore size distribution had significant impact on SOC content. The TOC and MAOC contents were
both positively correlated with pores of <15 μm and of > 80 μm (P< 0.05) but negatively correlated with pores of 15-
30 μm (P< 0.05).
RDA was used to explain the relationship between the pore parameters and SOC fractions during the seasonal
FT process (Fig. 7). In the freezing period, a total of 53.29% of the SOC variation could be explained by pore
characteristics (Fig. 7a). Pore equivalent diameter had a significant impact on SOC content (P<0.05). In thawing
period, 52.90% of the SOC variation, with 50.99% on Axis 1 and 1.91% on Axis 2, was explained by pore
characteristics (Fig. 7b). Pore surface area and EqD played important roles in SOC dynamics of aggregates (P<0.05).



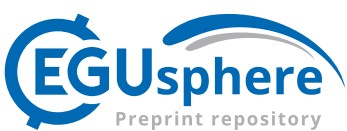

Table 2 Correlations between SOC content, soil microbial characteristics and soil structure of soil aggregates in freezing period and thawing period

**Freezing period**

| | Porosity | Equivalent diameter | Mean volume | Pore surface area density | Pore length density | Pore shape factor | Pd<15 | Pd15-30 | Pd30-80 | Pd>80 |
|---|---|---|---|---|---|---|---|---|---|---|
| TOC | 0.428 | -0.404 | -0.124 | 0.553 | **0.718*** | 0.241 | 0.420 | 0.084 | 0.316 | -0.235 |
| POC | 0.222 | -0.252 | 0.188 | 0.339 | 0.397 | 0.032 | **0.639*** | 0.123 | 0.410 | -0.273 |
| MAOC | 0.529 | -0.443 | -0.479 | **0.622*** | **0.865*** | 0.422 | 0.013 | 0.010 | 0.086 | -0.106 |

**Thawing period**

| | Porosity | Equivalent diameter | Mean volume | Pore surface area density | Pore length density | Pore shape factor | Pd<15 | Pd15-30 | Pd30-80 | Pd>80 |
|---|---|---|---|---|---|---|---|---|---|---|
| TOC | 0.582 | -0.507 | -0.036 | 0.326 | 0.396 | 0.199 | **0.811*** | **-0.834*** | -0.503 | **0.733*** |
| POC | 0.521 | -0.214 | -0.274 | 0.178 | 0.428 | 0.538 | 0.458 | -0.353 | -0.146 | 0.295 |
| MAOC | 0.409 | -0.498 | 0.117 | 0.296 | 0.234 | 0.071 | **0.727*** | **-0.818*** | -0.532 | **0.727*** |

Note: * represents the correlation is significant (P<0.05). Pd<15: volume percentage of pores <15 μm, Pd15-30: volume percentage of pores 15-30 μm; Pd30-80: volume percentage
of pores 30-80 μm; Pd>80: volume percentage of pores >80 μm.





Fig. 7 RDA analysis between SOC content and pore characteristics of aggregates in (a) the freezing period and (b) the thawing period.

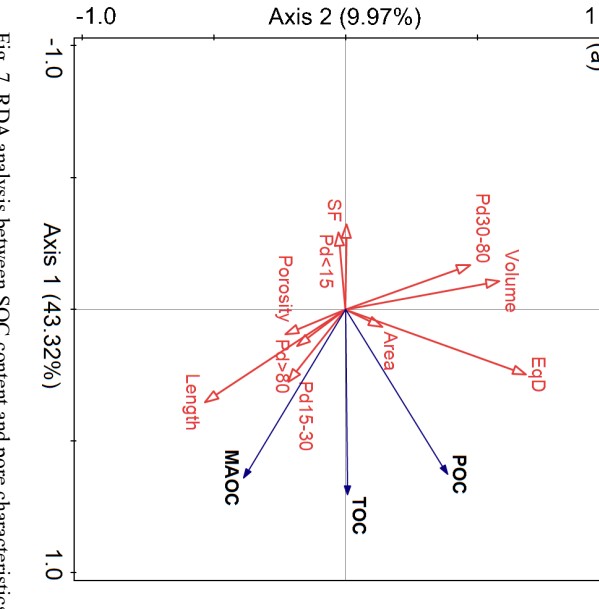

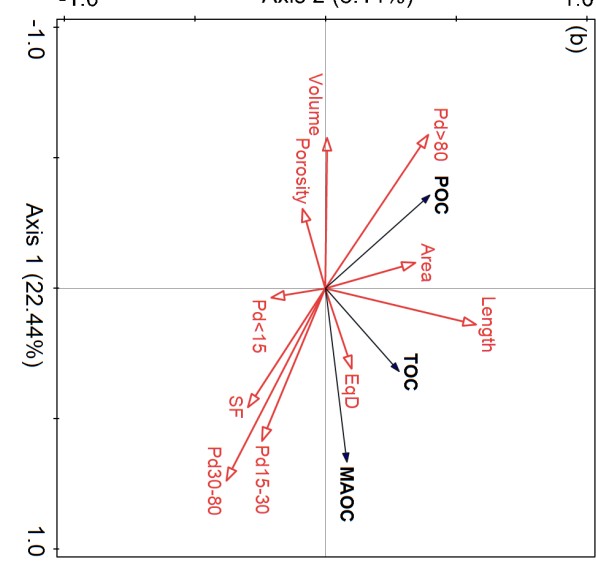

Note: Volume-pore volume, EqD-equivalent diameter of pores, Pd30-80-pores with diameter of 30-80 μm, SF-pore shape factor, Pd<15: pores with diameter of <15 μm, Pd15-30-

pores with diameter of 15-30 μm, Pd>80- pores with diameter of > 80 μm.



## 4. Discussion


Our results demonstrated that the volume percentage of aggregates was high in the stable frozen period. This
finding is consistent with previous results, which showed that FTCs resulted in an increase in macroporosity, with a
significant increase observed after the 3rd FTC compared to no FT control (Wu and Hu, 2024). The impact of seasonal
FT processes on aggregate pore structures is similar to the those of simulated FT cycles. The FT processes initially
altered soil pore network due to frost heave and reorganization of particles (Zhang et al., 2016). These changes
therefore drive soil development processes (Ping et al., 2015). Ma et al. (2020) found volume percentage of pores of >
100 μm in aggregates increased from 62.39% to 96.53% after 20 times FT cycles. During the freezing process, pore-
scale heterogeneities cause pressure gradients and the seepage of water from smaller to larger pores (Rempel and vam
Alst, 2013), and this process enhance the expansion of force heave (Skvortsova et al., 2018). Freezing could also
increase pore size by forming new connections among adjacent pores (Ma et al., 2020). The increase in pore size and
porosity could loosen the aggregate stability and increase pore air content, thus increasing the air pressure and
enhancing expansion (Lugato et al., 2010; de Jesus Arrieta Baldovino et al., 2021). We also found that the seasonal
FT process mainly affects the pore characteristics of 0.25-2 mm aggregates rather than those of > 2 mm aggregates,
especially in the pore surface area density and length density. Zhao and Hu (2023a) reported a similar significant
change in pore surface area density of 0.25-1 mm aggregates after FT cycles. Changes in surface area density and pore
length density or pores might be associated with pore shape (Rooney et al., 2022). In the freezing period, the frost
heave force of water is anisotropic, which increases the pore length and decreases the surface area. In summary,
freezing increased the pore volume and the impact of seasonal FT processes on pore characteristics is dependent on
aggregate size.
Our results revealed that the contents of SOC fractions were all high in the stable frozen period and low in the
unstable thawing period, which is similar to previous findings. Huang et al. (2021) found that the TOC content of
aggregates was high in January and February and showed a significant decrease in March due to FT processes. Many
studies have also reported the SOC loss at the beginning of the thawing period at regional scales (Song et al., 2014;
Song et al., 2020). This phenomenon can be explained by litter accumulation and suppressed microbial activities in
freezing periods (Han et al., 2018), as well as the aerobic environment intensifying SOC mineralization during thawing
(Liu et al., 2018; Liu et al., 2021). Additionally, in these two ecosystems, a large amount of plant litter accumulated
and transformed into substrates in the freezing period and served as important source of SOC. This phenomenon may
have occurred because soil aggregates are sensitive to physical disruption, and FTCs affect the stability of aggregates,
leading to a release of POC and MAOC that is trapped within aggregates by physical occlusion and facilitating



organic-mineral bonding between clay and microbial residues (Bailey et al., 2019; Six et al., 2004). So, the freezing
process promoted SOC accumulation while the thawing process induced a loss of SOC.

The results revealed that pore equivalent diameter explained most in the SOC variations. In the freezing process,

pores of < 15 μm served as preferential spots for POC stabilization. As the period is featured by SOC accumulation,
< 15 μm pores reduced SOC decomposition via limiting microbial access, gas diffusion and water availability, shifting
microbial metabolism to less efficient anaerobic respiration (Strong et al., 2004; Keiluweit et al., 2017; Wang and Hu,
2023). In the thawing period, the TOC and MAOC contents were both positively correlated with pores of > 80 μm
and negatively correlated with pores of 15-30 μm. Pores of > 80 μm serve as primary sites for residue entry and are
promoted by microbial materials and SOC, which enhance soil aggregation and thus drive more SOC to be protected
(Ananyeva et al., 2013; Dal Ferro et al., 2014; Zhang et al., 2023). Pores of 15-30 μm, on the other side, might have
negative impact on SOC protection (Liang et al., 2019). These pores are accompanied by enhanced microbial activities
led to the mineralization of greater amounts of SOC (Kravchenko et al., 2015; Zhang et al., 2023). In summary,
considering the dramatic change in porosity of <15 μm and >80 μm pores, the seasonal FT process altered the SOC
protection of aggregates via regulating pore size distribution.

In this study, we explored changes in the pore structure and SOC fractions of alpine soil macroaggregates in the

seasonal FT process. However, we could not isolate the impact of FT processes on soil structure and functions as
impacts from vegetation and climate could not be avoided under field conditions. Therefore, it is necessary to compare
the results based on indoor FT simulations and field sampling in future studies to clarify the importance of FT
processes in shaping pore structure and affecting soil functions. Recent studies have clarified the importance of
minerals (e.g., Fe, Al, and their oxides) in microscale SOC protection (Kang et al., 2024; Wang et al., 2024; Zhu et al.,
2024). Establishing relationships between structural characteristics and minerals can help us better understand the role
of pore structure in SOC protection mechanisms.
**5. Conclusion**

The findings of the study revealed that seasonal freeze–thaw processes regulate pore structure, and SOC

concentration of aggregates. The seasonal FT process significantly affected the pore surface area density and length
density of 0.25-2 mm aggregates. The freezing process promoted the formation of pores > 80 μm while thawing led
to shrinkage of pore space. Freezing enhanced the accumulation of SOC of aggregates and the more uniform
distribution of SOC among different soil layers. Thawing processes witnessed the loss of SOC. The seasonal FT
process altered the SOC protection of aggregates via regulating pore size distribution. In the freezing process, pores
of < 15 μm enhanced the protection of SOC of aggregates by limiting microbial access and shaping anerobic




environments. In the thawing process, pores of 15-30 μm contributed to the SOC loss. Overall, our study explains the
changes in SOC during the freeze-thaw process by innovatively establishing a pathway of FT-pore structure-SOC.
This study has critical implications for predicting soil structural dynamics and reducing uncertainty in global carbon
cycle predictions under climate change.
**Abbreviations**
FT: freeze-thaw, FTC: freeze-thaw cycle, UFP: unstable freezing period, SFP: stable frozen period, UTP:
unstable thawing period, STP: stable thawed period, EqD: equivalent diameter of pores, SF: shape factor, LMA: large
macroaggregate, SMA: small macroaggregate, SOC: soil organic carbon, TOC: total organic carbon, POC: particulate
organic carbon, MAOC: mineral-associated organic carbon.
**Declarations**
**Acknowledgement**
This study was financially supported by the National Science Foundation of China (Grant number: 42371107)
and the Project Supported by State Key Laboratory of Earth Surface Processes and Resource Ecology (2022-TS-03).
**CRediT authorship contribution statement**
Ruizhe-Wang: Conceptualization; data curation; formal analysis; methodology; writing-original draft; writing-
review & editing. Xia Hu: Funding acquisition; investigation; project administration; supervision; writing-review &
editing.
**Availability of data and material**
All data generated or analyzed during this study are included in this published article [and its supplementary
information files.
**Declaration of competing interests**
The authors declare that they have no known competing financial interests or personal relationships that could
have appeared to influence the work reported in this paper.



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
