# Peer review of "Freeze-thaw processes correspond to the protection-loss of soil organic carbon through regulating pore structure of aggregates in alpine ecosystems"

_EGUsphere, 2024_

## Community Comment (CC1)

Dear referee

We highly appreciate your valuable comments on our manuscript. We have carefully considered your suggestions and made revisions of our manuscript. The replies are as follows:

General comments

1. This is an interesting and important research area and is a currently relevant study within the broad realm of soil carbon loss as a function of melting permafrost. However, I have several concerns with the study methods and therefore the interpretation of results. The very low sample size of nine for the comparisons between freezing and thaw period soils raises questions about the assertions of statistically significant differences, particularly when taking the large standard errors into account. The main concerns relate to making multiple comparisons without adjusting for those multiple comparisons that could be quelled if the data and code were provided.

Thanks for raising this meaningful question. In total, we scanned 144 aggregates. We have added the data in Line 169. We admit that more replicates can make our results more convincing, especially for units with strong heterogeneity like soil aggregates. However, considering the high cost of CT scanning in China, we have done our utmost to achieve a qualified sample size to ensure the credibility of the results. In the future, we will try to analyze more samples if possible.

2. The second concern is the soil density fraction method, which is an outdated method from the early 1990s that has proven to be an imprecise method of density separation compared to the more commonly used sodium polytungstate method. Sodium hexametaphosphate solutions can only achieve densities of up to about 1.2-1.4 g/cm3, whereas the commonly agreed upon densities for separating mineral fractions are 1.6 -1.85 g/cm3, which cannot be achieved usi ng sodium hexametaphosphate. Results relating to the density separation are therefore unreliable. I recommend removing results related to the density fractions and down-scoping this manuscript to focus on the seasonal differences in pore properties and TOC content, including the correlation table but not the RDA, which is redundant information.

We highly appreciate your insightful opinion. The density fraction method used in our study was referred from Marriott and Wander (2006), Chen et al. (2020) and Fu et al. (2023), which all used 5% (m/v) sodium hexametaphosphate. In our future studies, we will adopt your suggestion

and use the sodium polytungstate method.

To reduce redundant information, the RDA image has been moved into Supplementary information.

References:

Chen, J., Xiao, W., Zheng, C., Zhu, B. 2020. Nitrogen addition has contrasting effects on particulate and mineral-associated soil organic carbon in a subtropical forest. Soil Biology and Biochemistry 142, 107708. doi: 10.1016/j.soilbio.2020.107708.

Fu, C., Li, Y., Zeng, L., Tu, C., Wang, X., Ma, H., Xiao, L., Christie, P., Luo, Y., 2023. Climate and mineral accretion as drivers of mineral-associated and particulate organic matter accumulation in tidal wetland soils. Global Change Biology 30, e17070. doi: 10.1111/gcb.17070.

Marriott, E.E., Wander, M.M., 2006. Total and labile organic matter in organic and conventional farming systems. Soil Science Society of America Journal 70, 950-286. doi: 10.2136/sssaj2005.0241.

3. The introduction is lengthy and could be revised to include less ancillary information and grammatical structure could be improved throughout.

Many thanks for your valuable comment. We have made careful revisions throughout the introduction which demonstrate a better linkage between our goal and the background. We wish to submit the revised manuscript.

Specific Comments

1. SOC fractionation performed according to 1992 methods using sodium hexametaphosphate. This is an outdated method that should be retired in favor of using sodium polytungstate solutions for more precise density separation. Only in cases where the researcher is building on previous data to form long-term datasets would it still be appropriate to use sodium hexametaphosphate for comparability between studies.

We highly appreciate your insightful opinion. The density fraction method used in our study was referred from Marriott and Wander (2006), Chen et al. (2020) and Fu et al. (2023), which all used 5% (m/v) sodium hexametaphosphate. For recent related studies on the QTP, the sodium hexametaphosphate was also used by Pan et al. (2024), Gu et al. (2024), etc. In our future studies, we will adopt your suggestion and use the sodium polytungstate method.

References:

Gu, J., Yang, F., Song X., Yang, S., Zhang, G., 2024. Edaphic regulation of soil organic carbon fractions in the

mattic layer across the Qinghai-Tibetan Plateau. Science of the Total Environment 943, 173814. Doi: 10.1016/j.scitotenv.2024.173814.

Pan, Y., Ren, L., Huo, J., Xiang, X., Meng, D., Wang, Y., Yu, C., Liu, Y., Suo, J., Huang, Y., 2024. Soil geochemistry prevails over root functional traits in controlling soil organic carbon fractions of the alpine meadow on the Qinghai-Tibet Plateau, China. Catena 237, 107814. doi: 10.1016/j.catena.2024.107814.

2.  Figure 2 is excellent!

Thank you very much!

3.  Table 2 is labeled as correlations between SOC content, soil microbial characteristics. It seems that microbial characteristics is not meant to be included since none of the variables presented fit that category.

We apologize for our mistake. We have removed "soil microbial characteristics" from the title. The current title is "Correlations between SOC content and soil structure of soil aggregates in freezing period and thawing period".

4.  Actual p-values should be provided in the text instead of presenting them as p <0.05.

We highly appreciate your valuable suggestion. The actual p-values will be added in the text throughout the revised manuscript. For example, in Line 313-314, "The TOC and MAOC contents were both positively correlated with pore length density (P=0.045 and P=0.006, respectively)."

5.  Fig. 5: When conducting multiple comparisons with the low sample size of nine, caution must be taken in interpretation of results. Without seeing the data itself, it is difficult to assess the validity of these results, given the high variability and low sample size. It is likely that the proportion of significant results would be relatively low given the sample size and variability. Further scrutiny of the data and statistical tests is necessary.

Thanks for reminding us this crucial issue. We certainly hoped to analyze as many samples as possible, especially for aggregates, which are highly heterogeneous structural units. However, considering the high cost of CT scanning, we could only meet the standard of n=9 to assure convincing results. We will try to expand our sample number if possible in future studies.

6.  Table 2 and Fig. 7 effectively present the same information – that is the strength and direction

of correlation among different covariates, so only one of the two should be presented.

Many thanks for raising this question. To avoid the data redundancy, the Table 2 has been moved into the Supplementary information.

7. The supplementary data table should include standard error for each variable measured.

We highly appreciate your comment. The standard error of variables has been added in the Supplementary Tables, which can be seen as follows:

Supplementary Table 1. Basic soil physio-chemical properties

| Ecosystem | Soil depth (cm) | Bulk density (g/cm³) | Soil water content (%) | pH | Organic C (g/kg) | Total N (g/kg) | Particle size composition (%) | | |
|---|---|---|---|---|---|---|---|---|---|
| | | | | | | | clay | silt | sand |
| KPM (meadow) | 0-10 | 0.77±0.19b | 35.76±15.01 | 6.50±0.35 | 85.26±29.38a | 7.66±2.22a | 9.05±2.65 | 33.60±6.10 | 57.35±8.73 |
| | 10-30 | 1.00±0.17a | 32.00±20.68 | 6.49±0.19 | 67.12±20.49ab | 6.94±1.37ab | 10.65±3.74 | 35.83±9.05 | 53.52±12.64 |
| | 30-50 | 1.07±0.05a | 24.18±13.04 | 7.17±0.32 | 25.35±6.78b | 2.66±0.45b | 11.84±2.57 | 34.88±4.98 | 53.28±7.32 |
| PFS (shrubland) | 0-10 | 0.83±0.23 | 42.57±4.57a | 6.64±0.40 | 64.42±11.22a | 7.00±1.12a | 13.95±0.56 | 47.56±1.25 | 38.49±1.69 |
| | 10-30 | 0.81±0.15 | 32.40±8.70ab | 6.82±0.22 | 44.11±6.88ab | 4.30±0.90ab | 14.59±0.86 | 46.85±1.00 | 38.56±1.73 |
| | 30-50 | 0.96±0.15 | 22.82±0.50a | 7.31±0.37 | 36.44±7.06b | 3.38±0.53b | 15.05±1.80 | 47.44±3.80 | 37.50±5.58 |

**Note:** KPM-*Kobresia pygmaea* meadow; PFS- *Potentilla fruticosa* shrub. The properties were measured with samples taken in the unstable freezing period. All data is presented with standard error (n=3). Different lowercase letters denote significant difference between soil layers.

8. I would be happy to provide technical corrections for a revised version of the manuscript.

Thank you very much for your affirmation and valuable comments on our research. We hope to get your more detailed suggestions on our revised manuscript.

---

## Community Comment (CC2)

Dear referee

Many thanks for your valuable comments on our manuscript. We have carefully considered your suggestions and made corresponding revisions. We hope that we can upload the revised manuscript for your further review. The reply are as follows:

General Comments

1. This manuscript presents field data of soil aggregate pore structure and carbon content through an annual freeze-thaw cycle. The measurements appear to have been carefully executed, and demonstrate some trends throughout the year for both pore structure and carbon content. The work also demonstrates strong correlations between some pore structure observations and carbon cycling through the year. Most strikingly, POC and MAOC pools strongly are associated with different pore characteristics during the freezing and thawing seasons. The review of soil aggregate FT mechanics is quite extensive.

   Many thanks for acknowledging our work.

2. Despite extensive literature review, the manuscript struggles to contextualize its findings. Most importantly, the relationships presented are purely correlational, and are difficult to assume as causal. Protection is postulated as the driving mechanism for carbon protection, but the seasonal inputs and outputs are hardly mentioned. Additional drivers like mineralogy, hydrology, and FT intensity are also not discussed. The influence of these factors has already been described in another manuscript by the same authors, where soil water content was found to be a critical factor (https://doi.org/10.1016/j.catena.2023.107359). This highly related study should be more carefully introduced and discussed in the present work. Moreover, the broader significance of carbon protection in aggregate pores is not strongly established by the manuscript. For example, the study region is generously introduced in the introduction, but does not return in any of the results, discussion, or conclusions. The manuscript could also be improved by a smaller number of better integrated citations. Grammar and paragraph structure could be improved and streamlined throughout.

   We highly appreciate your comprehensive and meaningful comments on the manuscript. Based on the results on RDA and correlation analysis, we referred to previous studies to ensure

that our findings were not out of casualty, which can be seen in the Discussion section, e.g. "In the freezing period, pores of < 15 μm served as preferential spots for POC stabilization. As the period is featured by SOC accumulation, < 15 μm pores reduced SOC decomposition via limiting microbial access, gas diffusion and water availability, shifting microbial metabolism to less efficient anaerobic respiration (Strong et al., 2004; Keiluweit et al., 2017; Wang and Hu, 2023)."

We realized the significant impact of other soil factors, especially moisture content, on organic carbon. In other studies, we attempted to compare the impact of soil structure and other factors on SOC. But in this manuscript, we focus on comparing the contributions of different soil structural parameters to carbon fractions.

In our discussion part, we have linked our findings with our previous work, as well as the features of the QTP. For example, in Line 376-382: In the thawing period, pores of <15 μm inhibited the POC loss. Previous studies proved that these pores reduced SOC decomposition via limiting microbial access and shifting microbial metabolism to less efficient anaerobic respiration (Strong et al., 2004; Keiluweit et al., 2017). On the QTP, the positive impact of soil moisture on SOC protection has been revealed in both aggregate scale and landscape scale (Ma et al., 2022; Wang and Hu, 2023). The thawing process is accompanied by an increase in microbial activity and moisture availability, pores of <15 μm are able to hold water surrounding the soil particles (Kim et al., 2021).

To avoid grammatic mistakes and bad organized components, we have thoroughly modified our manuscript. We wish to submit the revised manuscript.

3. I would be interested to see a closer look at the data, with increased focus on the seasonal cycle, causality, and other driving factors. I think the value of the annual time series was not fully explored, and suggest that the analysis could look more carefully at the changes in each layer of each ecosystem over time, rather than aggregating all the soil layers and both ecosystems into the same statistical analysis. For the interesting data and contribution to understanding challenging soil processes, I recommend this manuscript to be reconsidered with revisions to the analysis, discussion, and contextualization of the findings.

We highly appreciate your insightful suggestions. Analyzing in each layers/ecosystems can indeed yield some interesting results, but in this study, as we focus on the relationship between

pores and SOC, more analysis can lead to inconsistent results, which is not conducive to revealing the relationship. In the future research, we would include more indicators to comprehensively reveal the organic carbon protection mechanisms of different soil layers.

Specific Comments

1. The author's previous work in the region should be more thoroughly described and integrated into the manuscript. Discussion of mineralogy, soil water content, and inter-aggregate porosity would all aid in the interpretation of your novel findings here.

   Thanks very much for the comments. The related findings have been described and analyzed in the Introduction and Discussion sections. For example, in Line 85-89 (in Introduction): Our previous studies have showed that, alpine meadow soil aggregates of the QTP had dense pore networks with many elongated pores in them due to frequent FT cycles (Zhao et al., 2020). For typical ecosystems on the QTP, the aggregate protection of SOC was promoted by pores of <15 µm by limiting microbial access and the process was most closely associated with soil moisture content (Wang and Hu, 2023).

2. The introduction and conclusion could be strengthened by removing extraneous detail, while focusing more on the implications of the work. Climate change and the QTP is a very interesting topic, and the reader would be interested in the implications of your work to understanding the future of the region.

   Many thanks for the insightful comment. We have added some background and implications in our manuscript. For example, in Line 89-91: "Aggregate stability has been proved to impact SOC protection on the QTP and thawing-induced SOC loss of aggregates will translate into carbon emissions from the meadow to the atmosphere and exacerbate global warming (Ozlu and Arriga, 2021). Also in Line 399-401: Future research needs to further quantify the impact of soil structure on organic carbon, which will enable us to apply the mechanisms we have discovered to landscape scales to improve existing global carbon cycle predictions.

3. The data on vertical structure (eg Table 1) has potential to be interesting, but is largely

unsupported by the manuscript. I suggest it should either be presented with supporting discussion, or trimmed from the manuscript.

Thanks very much for the valuable comment. We have added the related discussion: Freezing also resulted in a more uniform distribution of SOC across different soil layers. This finding is consistent with the findings of Zhao and Hu (2023), which proposed the buffered difference in microbial biomass between soil horizons in the frozen period. These indicated the positive effect of freezing on vertical nutrient transport, which lacks investigations so far.

4. Table 2 and Figure 7 present some interesting correlations, but I would be interested to see a scatter plot (perhaps color-coded by ecosystem) for some of the key relationships. I'm worried that the seasonal differences reflect different ecosystem behaviors, rather than mechanistic causality.

Many thanks for your valuable suggestions. We have added the scatter plots of some crucial correlations as is shown in Fig. 7 and Fig. 8 and checked whether they reflected ecosystem behaviors. All data were presented as the mean value of each ecosystem in each FT period to avoid error caused by extreme values.

[Figure]

Fig. 7. Scatter plots of relationships between (a) SOC content and 15-30 μm pores and (b) SOC content and > 80 μm pores in the freezing process.

[Figure]

Fig. 8. Scatter plots of relationships between (a) TOC content and pore length density, (b) MAOC content and pore length density and (c) POC content and < 15 μm pores in the thawing process.

5. The results throughout the paper are presented without much discussion of the physical mechanisms. I think the results in changing pore structure would be much more compelling with thoughtful discussion of the physical mechanisms. The same goes for the mechanisms of carbon protection, considering the sources and sinks of carbon.

We highly appreciate your insightful comments. We have added the discussion of the physical mechanisms, which can be seen in Line 367-389: In the freezing period, pores of 15-30 μm had negative impact on SOC protection, this was consistent with our previous results (Wang and Hu, 2023). Pores of 15–30 μm are probably suitable habitat for soil microbes and support their activity, where greater SOC decomposition takes place (Kravchenko & Guber, 2017; Liang et al., 2019). Pores of >80 μm favoured SOC protection of aggregates. As the period was featured by SOC accumulation (especially residue entry), Pores of > 80 μm serve as primary sites for residue entry and are promoted by microbial materials and SOC, which enhance soil aggregation and thus drive more SOC to be protected (Ananyeva et al., 2013; Dal Ferro et al., 2014; Zhang et al., 2023). Freezing promoted the formation of these pores which were conducive to organic matter entry into aggregates. In the thawing period, pores of <15 μm inhibited the POC loss. Previous studies proved that these pores reduced SOC decomposition via limiting microbial access and shifting microbial metabolism to less efficient anaerobic respiration (Strong et al., 2004; Keiluweit et al., 2017). On the QTP, the positive impact of soil moisture on SOC protection has been revealed in both aggregate scale and landscape scale (Ma et al., 2022; Wang and Hu, 2023). The thawing process is accompanied by an increase in microbial activity and moisture availability, pores of <15 μm are able to hold water surrounding the soil particles (Kim et al., 2021). Therefore, POC associated

with these pores was less vulnerable to microbial processing and desorption due to equilibration with the more frequently exchanged soil solution (Schluter et al., 2022). The protection promotes the consequent transport of POC towards mineral sorption and thus contributes to the long-term SOC storage (Vedere et al., 2020). Overall, the FT-induced pore structure posed a positive impact on SOC protection in that: pores of > 80 μm promoted by freezing serve as primary sites for organic matter entry, while pores of <15 μm promoted by thawing inhibited POC decomposition through holding moisture.

6. Better paragraph structure and organization will improve the overall clarity and readability tremendously. I would be happy to provide more detailed comments on a revised manuscript.

Thanks very much for your insightful opinions. We have improved the organization especially for the Introduction and Discussion sections in the revised manuscript, which will present a better link between our findings and backgrounds of the study.

Overall, thanks again for all your valuable comments.

---

## Author Response (AR2)

Dear editor

Many thanks for raising the problems and suggestions in our manuscript (ID: egusphere-2024-1833). The comments are helpful to the improvement of our paper, and have been incorporated into the revised version of the manuscript. Our responses to the comments are listed below:

**Referee#1**

Thank you to the authors for their revisions, which improved the clarity and quality of the manuscript. The presentation of the methods is quite clear for the most part. The presentation of the data has also been appreciably streamlined. The introduction and especially discussion have both been strengthened in organization and clarity. There are still some areas where I think would be beneficial to clarify, mostly in the discussion section.

Many thanks for acknowledging our revisions and offering valuable suggestions.

Specific topics:

1. The analysis consistently differentiates between meadow and shrubland, or example figures 5,6 and 7. However, the differences between meadow and shrubland are not discussed. Why not either group the data, or discuss the differences? More subtly, looking at meadows vs shrublands, this highlights the difficult of interpreting differences - random heterogeneity, or a systematic signal?

Thanks for raising the valuable question. Soils of the meadow and shrubland ecosystems both belongs to Gelic Cambisols according to the FAO UNESCO system (IUSS Working Group WRB, 2022). These two ecosystems are representative of typical alpine ecosystems on the QTP. These two ecosystems also both have *Kobresia* vegetation and mattic epipedon. The objectives of the study were to quantify changes in pore structure and SOC fraction contents of aggregates in typical alpine ecosystems during the seasonal FT process, and to find the pattern of the effects of FT on pore structure and SOC fraction contents of aggregates as well as their relationships. So, we did not focus on the differences between meadow and shrubland. In the future, we will focus on the differences between these two ecosystems.

2. On a related topic to meadows/shrubs, I am wondering about the difference between pore processes in different size aggregates. Is there any reason to expect different processes in differently sized aggregates, or is this just random heterogeneity? If the processes should be the same between different sized aggregates, perhaps it is fraught to interpret differences between aggregate sizes (eg discussion starting at line 346).

Thanks very much for your valuable question. Soil aggregation processes can be explained by the aggregate hierarchy theory (Tisdall and Oades 1982). Briefly, primary particles and silt-sized aggregates are first bound together into microaggregates by persistent binding agents (e.g. humus, disordered aluminosilicates). Then, several small-sized aggregates are bound together into lager ones by temporary and transient organic binding agents (e.g. fungal hyphae, roots) (Six et al., 2004). Therefore, there are differences in the internal adhesion of aggregates of different sizes. Previous studies have proved that cementing agents (e.g. organic matter and metallic oxide) could affect intra-aggregate pore structure due to its influence on the morphological characteristics, permutation, and the combination of particles (Schweizer et al., 2019; Peng et al., 2022).

Soil aggregates of >0.25-2 mm and >2 mm are both crucial units for SOC protection. In alpine ecosystems, soil pores were formed and developed by complex interactions among root penetration (Hu et al., 2020), FT processes (Zhao et al., 2020), and microbial/animal activities, etc. Considering the differences in stability and internal binding between the two types of aggregates, investigating their pore network can help better evaluate their carbon protection ability.

However, it is pitiful that previous studies have not investigated the pore formation processes of different aggregates. We hope to conduct further investigations in the future.

References:

Hu, X., Li, X., Li, Z., Gao, Z., Wu, X., Wang, P., Lyu, Y., Liu, L.: Linking 3-D soil macropores and root architecture to near saturated hydraulic conductivity of typical meadow soil types in the Qinghai Lake Watershed, northeastern Qinghai-Tibet Plateau. Catena, 185, 104287. https://doi.org/10.1016/j.catena.2019.104287, 2020.

Peng, J., Wu, X., Ni, S., Wang, J., Song, Y., Cai, C.: Investigating intra-aggregate microstructure characteristics and influencing factors of six soil types along a climatic gradient. Catena 210, 105867. https://doi.org/10.1016/j.catena.2021.105867, 2022.

Schweizer, S.A., Bucka, F.B., Graf-Rosenfellner, M., Kogel-Knabner, I.: Soil microaggregate size composition and organic matter distribution as affected by clay content. Geoderma, 355, 113901. https://doi.org/10.1016/j.geoderma.2019.113901, 2019.

Six, J., Bossuyt, H., Degryze, S., Denef, K.: A history of research on the link between (micro)aggregates, soil

biota, and soil organic matter dynamics. Soil Tillage Res., 79, 7-31. https://doi.org/10.1016/j.still.2004.03.008, 2004.

Tisdall, J.M., Oades, J.M.: Organic matter and water-stable aggregates in soils. J. Soil. Sci., 33, 14-163. https://doi.org/10.1111/j.1365-2389.1982.tb01755.x, 1982.

Zhao, Y., Hu, X., Li, X.: Analysis of the intra-aggregate pore structures in three soil types using X-ray computed tomography. Catena, 193, 104622. https://doi.org/10.1016/j.catena.2020.104622, 2020.

3. The discussion of vertical Coefficient of Variation is interesting, but possibly also a good example of correlation vs causation. The manuscript shows very nicely that there are seasonal changes in vertical TOC distribution, but I am skeptical that this is driven by FT processes altering pore structure. It seems more likely that vertical carbon structure and pore structure are both being driven by seasonal changes in hydrology, phenology, and temperature.

We highly appreciate your insightful comment. We agreed with your perspective that the vertical SOC distribution was driven by multiple factors including freeze-thaw, phenology and hydrology, etc., rather than solely FT processes. To avoid ambiguity, we have modified the related discussion into (Line 357-363): The freezing process was also accompanied by a more uniform distribution of SOC across different soil layers. This finding corresponds to Zhao and Hu (2023), which proposed that freezing buffered difference in microbial biomass between soil horizon. Apart from seasonal dynamics in phenology and hydrology, differences in external disturbances and SOC turnover rates from topsoil to deep soil also contributed to this phenomenon (Sun et al., 2020; Wang et al., 2022). Therefore, freezing might pose indirect and positive impact on vertical nutrient distribution, which lacks investigations so far.

4. I appreciate the mention of mineralogy and vegetation as potentially confounding factors, and I think that this discussion can go further. Several parts of the discussion make claims that appear more causal than is warranted, such as "Freezing also resulted in a more uniform distribution of SOC across different soil layers" on line 326-327. - Three is certainly a seasonal cycle, but it feels difficult to claim that freezing is the causal driver. Several instances like this can be improved by emphasizing correlation, not causation.

Many thanks for raising the insightful question. The discussion of mineralogy and vegetation has been added as is shown in Line 397-399: For example, the presence of iron-rich substances can hamper microbial degradation of organic compounds, and the Fe-OC accounted for

approximately 20% of the total carbon pool on the QTP (Mu et al., 2016). This mechanism can be closely associated with soil moisture and enzyme activities, both of which are altered by FT processes (Li et al., 2023; Hu et al, 2024). Also, in Line 380-382: Therefore, POC associated with these pores was less vulnerable to microbial processing and desorption as thawing enhanced exchanged soil solution and consequent equilibration (Schluter et al., 2022). We will focus more on soil minerology and vegetation on our following studies.

We highly agreed with you in that "it feels difficult to claim that freezing is the causal driver" and we have revised the expressions to emphasize correlations rather than causation throughout the manuscript. For example (Line 357-358), the freezing process was also accompanied by a more uniform distribution of SOC across different soil layers.

There are a small number of specific corrections, I would be happy to go through and give complete proofreading on the next revision.

Thanks again for your insightful comments. Thanks again for your insightful comments. We have made several revisions concerning the grammatical structure.

**Referee #2**

The authors have improved the original MS. version, and I thank them for their responses to my initial concerns. However, there are still substantial issues with the revised ms that I will outline below:

Regarding the low sample size and statistical significance, I can appreciate the difficulty and expense of including additional replicates, but my concerns about the reliability and interpretation of statistical significance remain. The authors have not responded to my comment about the data and code availability (they state at the end of the ms that the data are included with the published article) or the multiple comparisons made without statistical adjustment. The number of statistically significant differences discussed in the text are surprising given the apparent lack of differences in the bar plot means. For example, in Fig. 5a, the differences between shrubland >2 mm and shrubland 0.25-2 mm do not appear significant, given the wide SE. Similarly, Fig. 5f meadow >2 and meadow 0.25-2 mm do not appear significantly different. I therefore encourage the authors to double check their calculations.

We highly appreciate your insightful opinions. Due to the scanning accuracy of industrial CT, the sample size was limited. We really hope to expand the sample size to reach a more accurate assessment in future studies if possible.

We have carefully checked our calculations (especially the significant test) and we apologize for any errors existing in previous figures and tables. The revised results can be seen in Fig. 5. The corresponding descriptions were also revised, for example, in Line 252-254: The seasonal FT process did not alter the porosity, pore volume and EqD significantly (Fig. 5a, 5b and 5c). In both ecosystems, significant variations were found in the mean pore volume between >2 mm and 0.25-2 mm aggregates ($p<0.05$).

[Figure]

Fig. 5. Pore characteristics of soil aggregates during the seasonal FT process. (a) porosity, (b) pore equivalent diameter, (c) mean volume of pores, (d) pore surface area density, (e) pore length density and (f) pore shape factor. Bars represent the mean $\pm$ standard error (n=18). ** represents significant differences between pore characteristics in freezing period and thawing period ($p<0.05$). Different lowercase letters denote significant differences between pore characteristics of >2 mm aggregates and 0.25-2 mm aggregates ($p<0.05$).

Note: LMA->2 mm aggregates, SMA-0.25-2 mm aggregates, KPM-the meadow ecosystem, PFS-the shrubland ecosystem.

Specific comments

1. Regarding the below statements in the Abstract (L20-25; L27-30):"The total organic carbon (TOC), particulate organic carbon (POC) and mineral-associated organic carbon (MAOC) contents of aggregates were high in the stable frozen period and low in unstable thawing period, demonstrating that freezing process enhanced SOC accumulation while early stage of thawing led to SOC loss. In the freezing period, pore structure inhibited SOC loss by promoting the formation of >80 μm pores. In the thawing period, pores of <15 μm inhibited SOC loss. Our results revealed that changes in pore structure induced by FT processes could positively

contribute to SOC protection of aggregates." I have concerns about the extent to which the data support these assertions. The expanded discussion related to the mechanisms of SOC protection and loss in pores is helpful in the revised ms version, but the relationships between TOC, fraction C, and pore size still remain correlational rather than causational in my view. The wording in the abstract should be changed to reflect the speculative nature of these assertions. In future studies, soil respiration measurements and direct DOC measurements would be a more direct way to capture the losses of soil carbon, although admittedly difficult to capture in situ in freeze-thaw field conditions.

We highly appreciate your valuable comment. We have revised our expressions to emphasize correlations rather than causations. For example, in Line 23-25: demonstrating that freezing process were positively associated with SOC accumulation while early stage of thawing witnessed SOC loss. Also, in Line 26-30: In the freezing period, the SOC accumulation might be enhanced by the formation of >80 μm pores. In the thawing period, pores of <15μm was positively correlated with SOC concentration.

We strongly agree with your views on future research, which has been incorporated into the Discussion part (Line 393-395): Despite the difficulty in in-*situ* monitoring, soil respiration measurements and DOC measurements would be a more direct way to capture the loss pathways of SOC exerted by thawing.

2. L245-246: Since porosity reflects the proportion of soil volume that is made up of pore space (so both number and size of pores), it is probably more correct to state that thawing contributed to an increase in the number of pores of size <15 μm, instead of the porosity of them.

We highly appreciate your insightful suggestion. The related sentence has been modified in Line 239-241: The results showed that freezing process increased the proportions of pores of > 80 μm while thawing contributed to the increase in volume percentage of pores of <15 μm.

3. Fig 4. It would be helpful to again define acronyms in the figure caption.

Thanks very much. The acronyms have been added in the figure caption: Note: UFP-unstable freezing period, SFP-stable frozen period, UTP-unstable thawing period, STP-stable thawed period.

4. Figures 7 and 8 are good additions.

   Many thanks for acknowledging our related revisions.

5. As with the first MS version, there are still many opportunities for improvements in grammatical structure and sentence flow to improve readability, but it is more important that the substantial concerns be addressed first. I remain willing to address specific grammatical issues in subsequent versions.

   Thanks very much for your comments improved our work, and the comments are helpful to the improvement of our paper. We have made several revisions concerning the grammatical structure. For example, in Line 399-401: This mechanism can be closely associated with soil moisture and enzyme activities, both of which are altered by FT processes (Li et al., 2023; Hu et al, 2024). Thanks very much for your willingness to address these issues.

---

## Author Response (AR3)

Dear editor

We highly appreciate your recognizing our work and offering valuable suggestions.

We have thoroughly checked the Abstract and removed all the unnecessary abbreviations. For example (Line 13-14): However, the impact of seasonal freeze-thaw processes on pore structure and its impact on SOC fractions have been overlooked. Please check the revised manuscript.

Best regards,

Xia Hu, Ruizhe Wang